# Understanding stage of innovation of invasive procedures and devices: protocol for a systematic review and thematic analysis

Darren L Scroggie [1,2] Daisy Elliott [1] Sian Cousins,[1] Kerry NL Avery [1] Jane M Blazeby [1,2] Natalie S Blencowe [1,2]

[1]NIHR Bristol Biomedical Research Centre, Bristol, UK
[2]University Hospitals Bristol and Weston NHS Foundation Trust, Bristol, UK

**Correspondence to**
Dr Darren L Scroggie;
darren.scroggie@bristol.ac.uk

## ABSTRACT

**Introduction** Surgical innovation has generally occurred in an unstandardised manner. This has led to unnecessary exposure of patients to harm, research waste and inadequate evidence. The IDEAL (Idea, Development, Exploration, Assessment, Long-term follow-up) Collaboration provided a set of recommendations for evaluating surgical innovations based on their stage of innovation. Despite further refinements and guidance, adoption of the IDEAL recommendations has been slow; an important reason may be that determining the stage of innovation is often difficult. To facilitate evaluation of surgical innovations, there is a need for a detailed insight into what stage of innovation means, and how it can be determined. The aim of this study is to understand the concept of stage of innovation as reported in the literature.

**Methods and analysis** A systematic review is being conducted. Ovid MEDLINE and Embase databases were searched from their inception until July 2021 using an iteratively developed strategy based on the concepts of stage of innovation, invasive procedures or devices and guidance. Articles were included if they described an approach to evaluating surgical innovations in stages, described a method for determining stage of innovation, described indicators of stage of innovation, defined stages or described potential sources of stage-related information. Conference abstracts and non-English language articles were excluded. Other articles were detected from citations within included articles and suggestions from experts in surgical innovation. Data will be extracted regarding approaches to evaluating surgical innovations, methods for determining stage of innovation, indicators of stage of innovation, definitions of stages and potential sources of stage-related information. A thematic analysis will be conducted, and findings summarised in a narrative report.

**Ethics and dissemination** Ethical approval will not be required. This systematic review will be published in a peer-reviewed journal and presented at appropriate conferences.

**PROSPERO registration number** CRD42021270812.

## STRENGTHS AND LIMITATIONS OF THIS STUDY

⇒ A comprehensive search strategy and inclusive eligibility criteria will maximise the breadth of data obtained.
⇒ Thematic analysis is well-suited for searching for meaning and current understanding regarding abstract concepts such as stage of innovation.
⇒ An improved understanding of stage of innovation may support wider adoption of approaches to evaluating surgical innovations.
⇒ This study will be limited to articles written in the English language.

## INTRODUCTION

Progress in surgical practice has been driven by innovation. The continuous development and implementation of new ideas has led to the myriad surgical procedures in widespread use today. Innovation of surgical procedures has often been accompanied by innovation of related medical devices. For example, endoluminal techniques and deployable artificial heart valves have resulted in effective and safe treatment options for otherwise untreatable patient cohorts.[1] While innovation aims to improve outcomes, it is associated with unknown risks and potential harm.[2] Many deaths resulted from bold innovation efforts during the early years of cardiac surgery.[3] More recently, it became apparent that patients had experienced unanticipated complications following the use of new vaginal mesh procedures for stress urinary incontinence.[4] Initially the innovation was considered to be safe and effective; the potential for harm was realised after several years.

The governance processes for medical devices and invasive procedures are different in the UK. The regulation of medical devices is the responsibility of the government via the Medicines and Healthcare products Regulatory Agency (MHRA).[5] Products which meet the legal definition of a medical device may be subject to a conformity assessment by a UK approved body, which considers scientific

data, manufacturing processes and quality management. A UK Conformity Assessed certificate is issued if the checks are satisfactory. The National Institute for health and Care Excellence (NICE) considers the governance of new invasive procedures to be the responsibility of healthcare provider organisations.[6] NICE's Interventional Procedures Programme recommends that National Health Service providers should have governance structures to review, authorise and monitor the introduction of new procedures.[6] The Royal College of Surgeons of England (RCS) advocates the creation of local surgical innovation committees dedicated to this role.[7] These governance processes for new invasive procedures are not legal requirements, and are not enforced by the government. Furthermore, there has been a lack of clarity about when and how to implement governance mechanisms for new invasive procedures.[8]

Identifying when surgical innovation is occurring is often problematic, as the distinction between routine variation in surgical practice and innovation is not clear.[9] The lack of a clear definition of surgical innovation makes it difficult to understand when the governance processes for new procedures suggested by NICE and the RCS should be implemented. The Macquarie Surgical Innovation Identification Tool was developed to help surgeons and healthcare providers recognise innovation prospectively, although its usefulness has not yet been established.[8 10]

There has been increasing awareness of the need to improve the evaluation of surgical innovations.[11] Pharmacological innovation provides an appropriate comparator, as a well-defined phased approach for determining the safety and effectiveness of new drugs is widely used.[12] This approach minimises the risk of harm, while optimising the scientific quality of evidence.[12] Even beyond phased clinical trials, long-term risks, adverse events and rare outcomes associated with new drugs are monitored using the MHRA's 'yellow card' scheme.[13] No comparable approach has been widely applied to surgical innovation, which has been generally unstandardised, under-governed and marred by a lack of high-quality evaluation.[14 15]

The IDEAL (Idea, Development, Exploration, Assessment, Long-term study) Collaboration has promoted the concept of staged surgical innovation over the past decade. Its most recent iteration of the 'IDEAL framework' described a sequence of five stages of innovation of invasive procedures; a modification of the framework known as 'IDEAL-D' dealt with devices.[16 17] The IDEAL framework begins with the first use in living humans at the 'Idea' stage, and finishes with ongoing surveillance of the widely adopted and stable procedure during the fifth 'Long-term study' stage. There is also a preclinical stage, which is not part of the framework itself, representing innovation prior to the first use in a living human. The framework provides recommendations regarding evaluation at each stage, with the aim of improving the quality of research in surgery. Despite further refinement of the framework and publication of practical guidance, its adoption has been relatively slow, and many authors

attempting to use the framework have failed to correctly apply its principles.[18 19] An important reason may be that determining the stage of innovation of a procedure or device is often difficult.[19] To facilitate a methodical approach to evaluating surgical innovations, there is a need for a detailed insight into what stage of innovation means, and how it can be determined.

The aim of this study is to understand the concept of stage of innovation as reported in the literature. The objectives are: to identify existing approaches to evaluating surgical innovations in stages, including methods for determining stage of innovation; and to identify related problems, with solutions or suggestions for further development.

## METHODS AND ANALYSIS

A systematic review will be conducted and reported in accordance with the recommendations of the PRISMA (Preferred Reporting Items for Systematic reviews and Meta-Analyses) statement.[20] This protocol has been reported in accordance with the PRISMA-Protocols statement (see online supplemental file 1).[21] Any subsequent modifications will be described in the final report. We anticipate completing the analysis by June 2022. This systematic review is registered with PROSPERO.

### Search strategy

Published literature was searched using the Ovid MEDLINE and Embase online bibliographic databases. The search was developed with the assistance of an information specialist, using scoping searches to iteratively refine the strategy. The broad search concepts were stage of innovation, invasive procedures or devices and guidance. The search strategies for Ovid MEDLINE and Embase are shown in online supplemental files 2 and 3. The sensitivity of the search strategy was verified by checking its ability to detect original reports from the IDEAL Collaboration, and refining the strategy as required to ensure they were detected.[16 22] Reference lists within selected articles from the search results were screened for other relevant articles. Experts in surgical innovation will be asked to suggest other relevant articles which may not have been detected by the search strategy.

### Eligibility criteria

A search result was eligible for inclusion if it: (i) described an approach to evaluating surgical innovations in stages, or; (ii) described a problem with an approach, proposed a solution or made a recommendation for further development, or; (iii) described a method for determining the stage of innovation of an IP/D, or; (iv) described a problem with a method for determining stage of innovation, proposed a solution or made a recommendation for further development, or; (v) described a property of an innovation which can infer its stage of innovation, or; (vi) defined a discrete stage of innovation, or; (vii) described a potential source of stage-related information. This included, but was not

limited to, the concept of stage of innovation as described by the IDEAL Collaboration.[16] Only innovation of IP/Ds was eligible; innovation relating to other forms of therapy, such as pharmaceuticals, was excluded.

Invasive procedures were defined according to the definition proposed by Cousins *et al*:

> An invasive procedure is one where purposeful/deliberate access to the body is gained via an incision, percutaneous puncture, where instrumentation is used in addition to the puncture needle, or instrumentation via a natural orifice. It begins when entry to the body is gained and ends when the instrument is removed, and/or the skin is closed. Invasive procedures are performed by trained healthcare professionals using instruments, which include, but are not limited to, endoscopes, catheters, scalpels, scissors, devices and tubes. Where invasive procedures also involve the administration of a medicinal product, these could be categorised as being part of an 'invasive procedure' when operator skill is required for its administration within the body, that is, when an internal action is performed to administer the product or the product is administered to a targeted anatomical area … There are also procedures which involve operator skill to target something inside the body (eg, electromagnetic radiation in the eye) without an incision, percutaneous puncture or instrumentation via a natural orifice. These types of procedures do not fall within the definition of an invasive procedure.[23]

Devices were restricted to a subset of medical devices as defined by the Medical Devices Regulations 2002 (SI 2002 No 618, as amended): general medical devices or active implantable medical devices which were both invasive and therapeutic.[24] This excluded diagnostic medical devices. Approaches to surgical innovation included descriptions, frameworks, models, classifications, typologies, guidelines, policies or recommendations, which were intended to introduce structure into the process of evaluating surgical innovation. A wide range of article types were eligible, including reports of primary and secondary research, guidelines and opinion pieces. There were no restrictions by date of publication. Conference abstracts and articles presented in languages other than English were excluded because of the difficulties associated with data extraction.

### Screening

Search results were imported to EndNote X9 reference management software (Clarivate, 2013). Duplicate search results were resolved using EndNote. Search results from published literature were first screened by title and abstract. Full-text articles were then retrieved, read in full and screened. Screening of each title, abstract and full text was undertaken by two independent assessors, with conflicts resolved by discussion. A third assessor was consulted if a consensus was not reached, deciding the outcome by the majority opinion. The Rayyan web application was used to facilitate screening.[25] Citations within

eligible full-text articles were checked for any further potentially relevant articles.

### Data extraction

Data extraction will be facilitated using the REDCap electronic data capture tool.[26] A data collection form will be predefined, piloted and iteratively refined using samples of articles returned by the search. Included articles will be read in full to identify relevant data, which may be extracted from any sections of the articles. Data will be extracted verbatim where possible.

Data will be extracted regarding the following:
► Article characteristics (eg, title; year; authorship; type of medium; funding; country).
► Article type (eg, letter; narrative review).
► Approaches to evaluating surgical innovations (eg, name of approach; original citation; form of presentation of approach; how it is used; appraisals, evaluations, opinions, critiques, recommendations, suggestions or comments about the approach).
► Stage of innovation (eg, meaning of stage of innovation; definitions of discrete stages; methods, guidance, advice, recommendations, descriptions, instructions or explanations regarding how to determine stage of innovation; appraisals, evaluations, opinions, critiques, recommendations, suggestions or comments regarding any process for determining stage of innovation; properties or characteristics which might infer stage of innovation).
► Sources of stage-related information (eg, where information about stage of innovation might be obtained; appraisals, evaluations, opinions, critiques, recommendations, suggestions or comments about an information source; which properties or characteristics might be obtained from an information source; methods, guidance, advice, recommendations, descriptions, instructions or explanations regarding how information regarding stage of innovation might be obtained from an information source).
► General comments (ie, any other text which might be relevant to the aim of the study).

Data will be extracted independently from all eligible articles by both reviewers. The two sets of data will be compared for consistency; discrepancies will be resolved by discussion, or consultation with a third reviewer if required.

### Data analysis

Descriptive data will be summarised in a table, including article characteristics and article types. A thematic analysis will be conducted for data relating to approaches to evaluating surgical innovations, stages of innovation, sources of stage-related information and general comments, in accordance with the method described by Braun and Clarke.[27] Thematic analysis is appropriate for searching across a data set for repeated patterns of meaning.[27] It is well-suited for inductively extracting semantics and understanding regarding stage of innovation. The analysis comprises six steps: familiarisation with the data, generation of initial codes, searching

for themes, reviewing themes, definition and naming of themes and production of a report. The thematic analysis will be conducted by two reviewers, with frequent comparisons of their codes and themes. Assessments of risk of study bias or methodological quality of included articles will not be appropriate for this study.

## Patient and public involvement

This study will not directly involve patients or members of the public, due to its methodological focus.

## Ethics and dissemination

Ethical approval will not be required as this will be secondary research utilising publicly available data. This systematic review will be published in a peer-reviewed journal and presented at appropriate conferences.

**Acknowledgements** The authors wish to acknowledge the expert assistance provided by Catherine Borwick, University of Bristol, in developing the search strategy.

**Contributors** NSB and JMB conceived the initial idea for the study and initiated the study. DLS, DE, SC, KNLA, JMB and NSB iterated the idea for the study and designed the study protocol. DLS prepared the initial and final drafts of the manuscript. DE, SC, KNLA, NSB and JMB critically revised the manuscript. All authors approved the final version of the manuscript.

**Funding** This study was supported by the Medical Research Council [MR/S001751/1] and NIHR Biomedical Research Centre at University Hospitals Bristol and Weston NHS Foundation Trust and the University of Bristol.

**Competing interests** None declared.

**Patient consent for publication** Not applicable.

**Provenance and peer review** Not commissioned; externally peer reviewed.

**ORCID iDs**
Darren L Scroggie http://orcid.org/0000-0002-5472-2602
Daisy Elliott http://orcid.org/0000-0001-8143-9549
Kerry NL Avery http://orcid.org/0000-0001-5477-2418
Jane M Blazeby http://orcid.org/0000-0002-3354-3330
Natalie S Blencowe http://orcid.org/0000-0002-6111-2175

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
