## [Reviewer comments · BMJ Open]

ARTICLE DETAILS

TITLE (PROVISIONAL)	Understanding stage of innovation of invasive procedures and devices: protocol for a systematic review and thematic analysis
AUTHORS	Scroggie, Darren; Elliott, Daisy; Cousins, Sian; Avery, Kerry; Blazeby, Jane; Blencowe, Natalie

VERSION 1 – REVIEW

REVIEWER	S. J. Tate Cardiff University
REVIEW RETURNED	05-Nov-2021

GENERAL COMMENTS	Thank you for the opportunity to review, I was interested to read this protocol for a systematic review investigating how the concept of the stage of surgical innovation is used in the literature. Overall, I thought the protocol paper was well written, and presented an idea for a review that was novel. I also believe that examining the conduct of research in surgery is useful. As highlighted by the authors, there are recent examples where more robust processes in the development and introduction of an innovative procedure might have prevented harm to patients. Thus efforts to improve research processes and innovation in surgery are important. I have a few specific questions/comments outlined below: Summary - you have Introduction - You have compared the stages of drug development to the stages of a new surgical innovation. A clear difference between the two is the regulatory framework that is applied as an innovation moves into clinical practice. You have not discussed the role of the bodies involved in the regulation of invasive procedures and associated devices. An introduction to the regulatory oversight would be useful to provide context. Methods - as per the point above, regulatory and public bodies such as the MHRA and NICE and professional organisations such as Colleges or Boards of surgery in different countries have guidance on how novel procedures should be approached and appraised. Will you try to identify and evaluate these?- you specifically state you will not include innovation relating to other forms of therapy, e.g. pharmaceuticals, but that you will use the definition of 'invasive procedure' suggested by Cousins et al which includes invasive procedures that administer a pharmaceutical product. Could you clarify if this means that a new
--

	procedure/technique/device combined with a therapeutic, for example TransArterial ChemoEmbolisation of a liver lesion will be included or not.
REVIEWER	Shiko Ben-Menahem ETH Zurich, Management, Technology, and Economics
REVIEW RETURNED	24-Nov-2021
GENERAL COMMENTS	The protocol seems adequate and well conceived. Missing is the date or timeline of the study. The description of the innovation process for pharmaceutical drugs seems superfluous. A reference to the changes in the medical device regulation (MDR) seems appropriate.

VERSION 1 – AUTHOR RESPONSE

Reviewer 1

1. **Reviewer comment:** *“Thank you for the opportunity to review, I was interested to read this protocol for a systematic review investigating how the concept of the stage of surgical innovation is used in the literature. Overall, I thought the protocol paper was well written, and presented an idea for a review that was novel. I also believe that examining the conduct of research in surgery is useful. As highlighted by the authors, there are recent examples where more robust processes in the development and introduction of an innovative procedure might have prevented harm to patients. Thus efforts to improve research processes and innovation in surgery are important.”*

Author response: We thank the reviewer for their support.

Introduction

2. **Reviewer comment:** “You have compared the stages of drug development to the stages of a new surgical innovation. A clear difference between the two is the regulatory framework that is applied as an innovation moves into clinical practice. You have not discussed the role of the bodies involved in the regulation of invasive procedures and associated devices. An introduction to the regulatory oversight would be useful to provide context.”

Author response: We thank the reviewer for this comment. We have replaced the description of clinical trial phases for new drugs with a description of how medical devices and new invasive procedures are governed in the UK. The following sentences have been added to the introduction: *“The governance processes for medical devices and invasive procedures are different in the United Kingdom (UK). The regulation of medical devices is the responsibility of the Government via the Medicines and Healthcare products Regulatory*

Agency (MHRA). Products which meet the legal definition of a medical device may be subject to a conformity assessment by a UK Approved Body, which considers scientific data, manufacturing processes and quality management. A UK Conformity Assessed (UKCA) certificate is issued if the checks are satisfactory. The National Institute for health and Care Excellence (NICE) considers the governance of new invasive procedures to be the responsibility of healthcare provider organisations. NICE's Interventional Procedures Programme (IPP) recommends that National Health Service (NHS) providers should have governance structures to review, authorise and monitor the introduction of new procedures. The Royal College of Surgeons of England (RCS) advocates the creation of local surgical innovation committees dedicated to this role. These governance processes for new procedures are not legal requirements, and are not enforced by the Government. Furthermore, there has been a lack of clarity about when and how to implement governance mechanisms for new invasive procedures.

Identifying when surgical innovation is occurring is often problematic, as the distinction between routine variation in surgical practice and innovation is not clear. The lack of a clear definition of surgical innovation makes it difficult to understand when the governance processes for new procedures suggested by NICE and the RCS should be implemented. The Macquarie Surgical Innovation Identification Tool (MSIIT) was developed to help surgeons and healthcare providers recognise innovation prospectively, although its usefulness has not yet been established."

Methods

- 3. Reviewer comment:** *"As per the point above, regulatory and public bodies such as the MHRA and NICE and professional organisations such as Colleges or Boards of surgery in different countries have guidance on how novel procedures should be approached and appraised. Will you try to identify and evaluate these?"*

Author response: The search strategy is designed to detect published articles which describe approaches to evaluating surgical innovations in stages. Any relevant approaches described by regulatory and public bodies are therefore expected to be identified.

Organisations such as MHRA and NICE have produced guidance relating to issues such as governance and ethics of new surgical procedures and devices; our study is primarily concerned with methods for evaluating surgical innovations, and therefore we would not include those guidance documents.

4. **Reviewer comment:** *“You specifically state you will not include innovation relating to other forms of therapy, e.g., pharmaceuticals, but that you will use the definition of ‘invasive procedure’ suggested by Cousins et al which includes invasive procedures that administer a pharmaceutical product. Could you clarify if this means that a new procedure/technique/device combined with a therapeutic, for example TransArterial ChemoEmbolisation of a liver lesion will be included or not.”*

Author response: Our study includes innovation relating to invasive procedures which involve the administration of a pharmaceutical, such as TransArterial ChemoEmbolisation (TACE) of liver lesions. This is in keeping with the definition proposed by Cousins *et al.* For clarity, we have added a quotation of their definition to the section on eligibility criteria:

“Invasive procedures were defined according to the definition proposed by Cousins et al: “An invasive procedure is one where purposeful/deliberate access to the body is gained via an incision, percutaneous puncture, where instrumentation is used in addition to the puncture needle, or instrumentation via a natural orifice. It begins when entry to the body is gained and ends when the instrument is removed, and/or the skin is closed. Invasive procedures are performed by trained healthcare professionals using instruments, which include, but are not limited to, endoscopes, catheters, scalpels, scissors, devices and tubes. Where invasive procedures also involve the administration of a medicinal product, these could be categorised as being part of an ‘invasive procedure’ when operator skill is required for its administration within the body, that is, when an internal action is performed to administer the product or the product is administered to a targeted anatomical area ... There are also procedures which involve operator skill to target something inside the body (eg, electromagnetic radiation in the eye) without an incision, percutaneous puncture or instrumentation via a natural orifice. These types of procedures do not fall within the definition of an invasive procedure.”

Reviewer 2

1. **Reviewer comment:** *“The protocol seems adequate and well-conceived.”*

Author response: We thank the reviewer for their support.

2. **Reviewer comment:** *“Missing is the date or timeline of the study.”*

Author response: We apologise for this omission. We have added the completion target to the methods section of the manuscript: *“We anticipate completing the analysis by June 2022.”*

3. **Reviewer comment:** *“The description of the innovation process for pharmaceutical drugs seems superfluous.”*

Author response: We have deleted the description of clinical trial phases for new drugs from the introduction.

4. **Reviewer comment:** *“A reference to the changes in the medical device regulation (MDR) seems appropriate.”*

Author response: We have added an explanation of how medical devices are regulated by the Medicines and Healthcare products Regulatory Agency (MHRA) in the UK, as detailed in our response above to Reviewer 1, Comment 2.

VERSION 2 – REVIEW

REVIEWER	S. J. Tate Cardiff University
REVIEW RETURNED	06-Jan-2022
GENERAL COMMENTS	I think the changes made have improved the clarity of the paper and I have no further comments. Many thanks.